# Cafe Robot: Integrated AI Skillset Based on Large Language Models

1st Jad Tarifi
*Integral AI*
Tokyo, Japan
jad@integral.ai

2nd Nima Asgharbeygi
*Integral AI*
Mountain View, USA
nima@integral.ai

3rd Shuhei Takamatsu
*Integral Japan*
Tokyo, Japan
shuhei@integral.ai

4th Masataka Goto
*Integral Japan*
Tokyo, Japan
goto@integral.ai

*Abstract*—The cafe robot engages in natural language interaction to receive orders and subsequently prepares coffee and cakes. Each action involved in making these items is executed using AI skills developed by Integral, including Integral Liquid Pouring, Integral Powder Scooping, and Integral Cutting. The dialogue for making coffee, as well as the coordination of each action based on the dialogue, is facilitated by the Integral Task Planner.

*Index Terms*—liquid pouring, powder scooping, cutting

## I. Introduction

Recently, the service robot market has become increasingly attractive due to advancements in AI technology. A proposal for an automated service robot system designed for use in cafes has emerged. This cafe robot is capable of taking orders from customers, preparing coffee and cakes, and serving them accordingly. The intricate actions required for coffee and cake preparation are accomplished through AI-based skills developed by Integral. LLM (Large Language Models)-based technology is employed to adapt the service in response to customer feedback and manage complex orders. The subsequent sections offer a comprehensive overview of these technologies and their practical demonstrations.

## II. System configuration

The setup comprises several devices, including two manipulation robots (Cobotta Pro and Cobotta), RealSense cameras attached to each robot's tips, a weight sensor, and a PC for controller operations. The cameras affixed to the tips of both robots serve various functions, such as cake recognition for cutting and powder recognition for scooping. The weight sensor is utilized for tasks involving liquid pouring and powder scooping. The primary robot, Cobotta Pro, features a gripper hand capable of grasping a variety of objects, including cups, spoons, kettles, pods, and spatulas. On the other hand, the tip of the secondary robot, Cobotta, is equipped with a fixed knife used specifically for cutting cakes.

The implemented setup enables natural language interaction with customers, allowing them to order cake and coffee seamlessly within the dialogue. Customers have the flexibility to specify their coffee preferences in terms of thickness, sugar content, and the amount of milk desired. Upon receiving an order, the system generates appropriate robot actions tailored to the specific order details. These actions are then executed, ensuring that customers receive their desired cake and coffee precisely as requested.

## III. Robot Skills

In the process of preparing coffee and cake, actions that cannot be accomplished through fixed robot movements are generated utilizing camera and weight sensor data, along with proprietary algorithms.

### A. Powder Scooping Skill

During the process of making coffee, a precise amount of coffee powder or sugar powder must be scooped from a container and poured into a cup. However, there are challenges associated with robot powder scooping. The powder can become unevenly distributed within the container, leading to interruptions in scooping the specific weight of powder required. This inefficiency results in the robot failing to scoop enough powder effectively. To address these challenges, the system employs unique recognition and motion generation technology.

Several random scoops of powder are conducted in advance for data collection purposes. During this process, data is recorded, including the distribution of the powder within the container as recognized by a camera, the specific locations from which the powder is scooped using the spoon, and the actual weight of the powder scooped. The inference model made by this data enables the inference of the appropriate locations for scooping based on the powder distribution.

This technology enables the robot to recognize the surface of the powder within the container and dynamically select strategies to overcome obstacles. These strategies may include gathering motions based on the distribution of powder in the container or scooping from an optimized target position. Through these optimized actions, the system achieves continuous and efficient scooping and pouring of powder.

### B. Liquid Pouring Skill

The process of making coffee necessitates pouring precise amounts of hot water or milk into the cup, presenting a challenge due to the diverse nature of liquids involved. Furthermore, recognizing liquids through computer vision poses additional difficulty.

To address these challenges, a simple system configuration comprising only weight sensors and robots is utilized. The action of pouring a random quantity of liquid is repeated numerous times for data collection purposes, during which the robot's movements and variations in the scale's readings are recorded. The end-to-end learned model made by this data is employed to generate pouring motions, enabling the robot to accurately pour a specified amount of liquid (measured by weight) into a target container with an error margin of within 1%. This system can be trained on various liquid types and viscosities, as well as different container types, enhancing its adaptability and versatility.

*C. Cutting Skill*

When serving cake, the robot must cut the cake to a specific width. However, as the position and shape of the cutting object (the cake) are indeterminate, it is necessary to dynamically determine the cutting pattern.

Utilizing computer vision, the cutting skill captures the position and shape of the cake, enabling the system to optimize the cutting plan while minimizing ingredient waste. This dynamic approach ensures precise and efficient cutting, tailored to the unique characteristics of each cake. Furthermore, this methodology is applicable not only to cakes but also to irregularly shaped objects, such as steak and tuna.

## IV. ORDERING SYSTEM

In constructing a robot system that integrates multiple algorithms, it typically involves complex action flows and state machines that consider various conditions. In the cafe system described here, the dialogue with the customer and the definition of robot behavior are realized based on LLM (Large Language Models).

During the dialogue with the customer, if the content pertains to an order, the system initiates the generation of robot behavior code. When generating robot behavior code based on natural language commands, if the information provided is insufficient, the system supplements it using a pre-run and runtime questioning mechanism. For instance, if the order is "black coffee," no further questioning is necessary, but if it is simply "coffee," the system prompts for additional details such as the amount of sugar and milk required.

Subsequently, the generated code undergoes simulation testing. If any errors are detected, the code is automatically corrected based on the error message to ensure practical functionality.

In summary, by utilizing LLM for customer dialogue and natural language information acquisition, and subsequently generating sequences of actions for the robot, the system enables autonomous task execution using other AI and skills in response to diverse orders.

## V. CONCLUSION

The cafe service robot, implemented utilizing the aforementioned setup configuration and robot skills, has been successfully demonstrated at various exhibitions. Throughout these exhibitions, the robot system has performed admirably, totaling nearly 500 successful demonstrations over the course of six days.

This showcase exemplifies the potential of future service robots, highlighting the advancements made possible through the evolution of AI.