# OpenReview forum: "Cafe Robot: Integrated AI Skillset Based on Large  Language Models"
_IEEE.org/2024/ICRA/Workshop/CookingRobot — CookingRobot2024 Poster_

### Official Review · Reviewer_QfgN · 2024-04-09
**The review for Cafe Robot: Integrated AI Skillset Based on Large Language Models**

**Rating:** 8
**Confidence:** 5

**Review:**

*Major Contribution:

This work shows a nice demonstration of making coffee according to the order by natural language, which includes powder scooping, liquid scooping, and cutting a cake. The authors share the real-world challenges.


*Video:

Great demonstration!
I am especially impressed by the use of the knife, which is not only for cutting but also for helping to insert the turner under the cake. I also liked the way of cutting the cake, which is inserting the knife and then shifting to the right a bit to separate the cake.
If there is no editing trim the video (e.g. picking or placing back the teaspoon, putting the cakes on the cutting board etc), it will increase more impact.


*Paper:

Describe the difficulty of each task in the real environment very well, which is important but usually not well-recognised by many researchers who haven’t tackled real-robot-cooking. Please discuss this deeply in the workshop!

*Major Comments:

The methods and techniques are not clear.

-	Which AI method did you use?
-	How to input to LLM model and command the robot?
-	How to use the camera and the weight sensor?
-	In the video, which part is autonomous? All motions are adjusted online with AI or some of them are predefined?

The advantage or uniqueness of your system is not clear.

-	The use of dialogue, shown in the example of “black coffee” and “coffee”, and error detection are very interesting, but how to realise the concept is not described.


*Minor comments:

Better to cite some relevant research or demonstrations.

---

### Official Review · Reviewer_QpCG · 2024-04-16
**The review for "Cafe Robot: Integrated AI Skillset Based on Large Language Models"**

**Rating:** 9
**Confidence:** 4

**Review:**

This paper realizes a cafe robot that makes coffee and cakes by integrating multiple AI-based skills and the LLM to manage them. Both AI-based skill execution and LLM-based management are effective methods for cooking robots. The fact that they are integrated as a system and have been demonstrated in real-world is also highly commendable. The content of this paper should elevate the discussion at the workshop to a higher level.

Major comments:
* It would be desirable to add more details about the used methods, such as what kind of models are used in the Powder Scooping Skill and Liquid Pouring Skill, in the Cutting Skill, how the cake is recognized using computer vision, how the cutting plan is optimized, and so on. How does the ORDERING SYSTEM with LLM realize the process of seeking additional information? These explanations would make the discussion in the workshop much more meaningful.
* A simple overview of the system or a diagram of the workflow of the cafe robot would also be very helpful.

Video:
* The video is very nice, stylish and the robot works well.